# Differential MicroRNA Expression in Porcine Endometrium Related to Spontaneous Embryo Loss during Early Pregnancy

**DOI:** 10.3390/ijms23158157

**Published:** 2022-07-24

**Authors:** Shengchen Gu, Xupeng Zang, Lei Jiang, Ting Gu, Fanming Meng, Sixiu Huang, Gengyuan Cai, Zicong Li, Zhenfang Wu, Linjun Hong

**Affiliations:** 1National Engineering Research Center for Breeding Swine Industry, College of Animal Science, South China Agricultural University, Guangzhou 510642, China; 15153918918@stu.scau.edu.cn (S.G.); xupeng_zang@stu.scau.edu.cn (X.Z.); jianglei@stu.scau.edu.cn (L.J.); tinggu@scau.edu.cn (T.G.); sxhuang815@scau.edu.cn (S.H.); cgy0415@scau.edu.cn (G.C.); lizicong@scau.edu.cn (Z.L.); 2State Key Laboratory of Livestock and Poultry Breeding, Guangdong Key Laboratory of Animal Breeding and Nutrition, Institute of Animal Science, Guangdong Academy of Agricultural Sciences, Guangzhou 510642, China; mengfanming@gdaas.cn; 3Guangdong Provincial Key Laboratory of Agri-Animal Genomics and Molecular Breeding, College of Animal Science, South China Agricultural University, Guangzhou 510642, China; 4Lingnan Guangdong Laboratory of Modern Agriculture, Guangzhou 510642, China; 5State Key Laboratory for Conservation and Utilization of Subtropical Agro-Bioresources, Guangzhou 510642, China

**Keywords:** spontaneous embryo loss, miRNA, implantation, endometrium, pig

## Abstract

Litter size is an important indicator to measure the production capacity of commercial pigs. Spontaneous embryo loss is an essential factor in determining sow litter size. In early pregnancy, spontaneous embryo loss in porcine is as high as 20–30% during embryo implantation. However, the specific molecular mechanism underlying spontaneous embryo loss at the end of embryo implantation remains unknown. Therefore, we comprehensively used small RNA sequencing technology, bioinformatics analysis, and molecular experiments to determine the microRNA (miRNA) expression profile in the healthy and arresting embryo implantation site of porcine endometrium on day of gestation (DG) 28. A total of 464 miRNAs were identified in arresting endometrium (AE) and healthy endometrium (HE), and 139 differentially expressed miRNAs (DEMs) were screened. We combined the mRNA sequencing dataset from the SRA database to predict the target genes of these miRNAs. A quantitative real-time PCR assay identified the expression levels of miRNAs and mRNAs. Gene Ontology and Kyoto Encyclopedia of Genes and Genomes pathway enrichment analyses were performed on differentially expressed target genes of DEMs, mainly enriched in epithelial development and amino acids metabolism-related pathways. We performed fluorescence in situ hybridization (FISH) and the dual-luciferase report gene assay to confirm miRNA and predicted target gene binding. miR-205 may inhibit its expression by combining 3′-untranslated regions (3′ UTR) of tubulointerstitial nephritis antigen-like 1 (*TINAGL1*). The resulting inhibition of angiogenesis in the maternal endometrium ultimately leads to the formation of arresting embryos during the implantation period. This study provides a reference for the effect of miRNA on the successful implantation of pig embryos in early gestation.

## 1. Introduction

Litter size is an essential economic indicator in the production and management of the pig industry [1]. Ovulation, sperm motility, fertilization rate, and embryo prenatal death affect the final number of offspring [2]. After fertilization, the main factor affecting litter size is the embryo’s prenatal death. The most significant embryo loss occurs during embryo implantation in early pregnancy; the embryo mortality rate is as high as 30% during the period between the day of gestation (DG) 10 and 30 [3]. Previous research has demonstrated that the main factor influencing spontaneous embryo loss in early pregnancy is abnormalities in mother–fetal communication. These interactions are controlled by the exchange of signals between the embryo and endometrium, such as: estrogen, microRNAs, extracellular vesicles, cytokines, chemokines, growth factors, mRNA destabilizing factors, and other substances. The failure of adjustment and coordination all contribute to spontaneous embryo loss. Insufficient blood supply to the uterus has been demonstrated to cause spontaneous embryo loss [3,4,5,6,7]. Previous studies found that vascular endothelium growth factors, hypoxia-inducible factor 1-α, IFN-γ, etc., all affect the angiogenesis between mother and fetus, ultimately leading to healthy and arresting embryos [8,9].

Small RNAs, including miRNAs, small interfering RNAs (siRNAs), and Piwi-interacting RNAs (piRNAs), play important roles in growth and development and genome integrity [10,11]. microRNA (miRNA) is known to regulate immune cell development and angiogenesis [12]. miRNA is a general term that refers to a class of RNA molecules of about 22 nt in length, endogenously expressed, conservative, and lack coding characteristics. The most commonly accepted mechanism is that at the post-transcriptional level, the seed sequence of miRNA is complementary to 3′-untranslated regions (3′ UTR) of mRNA, thereby inhibiting mRNA expression or degrading mRNA [13,14]. miRNAs mainly exist in various tissues and extracellular vesicles (EV) [14]. Numerous investigations in multiple species have indicated that miRNA is involved in embryonic development and in vitro oocyte maturation [14,15,16,17,18,19,20]. It has been shown that miR-181a and miR-181c are required for embryo implantation and placentation [13]. Previous research revealed that miRNAs in the endometrium are closely linked to immune response and angiogenesis-related genes [18]. As previously reported, miRNAs contribute to pregnancy establishment by influencing critical gene networks in immune cells, especially miR-233, miR-155, and miR-146b, which have immunomodulatory effects [21]. We selected sows at the end of the implantation period on DG 28 due to a higher percentage of spontaneous embryo loss. This work combines previously reported mRNA sequencing data with small RNA (sRNA) sequencing data in porcine endometrium at healthy and arresting embryo implantation sites on DG 28 [4]. Considering that miR-205 inhibits the expression levels of tubulointerstitial nephritis antigen-like 1 (*TINAGL1*), we inferred that miR-205 inhibits angiogenesis in maternal endometrial tissues, thereby reducing the nutrient supply to the embryo to inhibit the embryo development from the endometrial level; meanwhile, previous studies have shown that TINAGL1 is a positive regulator of angiogenesis that increases endothelial cell invasion and angiogenic sprouting As presented in Figure 1, we performed high-throughput sequencing, bioinformatic analysis, and experimental validation of arresting endometrium (AE) and healthy endometrium (HE) in sows on DG 28. The data is expected to provide a favorable reference for studying the molecular regulatory mechanism of spontaneous embryo loss at the end of embryo implantation.

## 2. Results

### 2.1. Analysis of miRNA Sequencing Results

Four samples per group were sequenced using the SE50 sequencing strategy. They were 55,191,250 and 57,050,149 raw reads obtained from porcine AE and HE tissue on DG 28, respectively. After excluding low-quality with 5′ adapter contaminant, without 3′ adapter and poly A/T/G/C reads, the two groups ended up with 54,855,390 and 56,720,713 clean reads. Clean reads account for 99.39% and 99.42% of raw reads in AE and HE on DG 28, respectively (Appendix A). Due to the length of animal sRNA ranging from 18 to 35 nt, and after 18–35 nt length screening of all clean reads, 53,821,520 and 55,755,655 reads were screened in AE and HE, respectively. miRNA is about 21–22 nt, and the length distribution of sRNA can assist us in determining the sRNA category; therefore, a miRNA length screening was performed (Figure 2A). The miRNA length was mainly distributed in 21, 22, and 23 nt, with 22 nt exhibiting the highest proportion. It was 36.86% and 32.49% in AE and HE on DG 28, respectively, indicating that 22 nt long miRNAs are a significant proportion of sRNA in the eight samples from two states. The category distribution of sRNA following length screening was determined by comparison to the reference sequence. The total matching rate in the two states is 98.17%.

These sRNA reads were compared to the miRBase database to determine the sequence, length, precursor sequence, occurrence frequency, and other related information of sRNA. In total, 52,785,605 and 54,780,288 sRNA reads were obtained from eight samples in AE and HE states, respectively. Furthermore, 38,967,952 (73.82%) and 41,204,005 (75.22%) reads were matched with known miRNAs in AE and HE sRNA libraries, and 590,471 (1.12%) and 492,504 (0.90%) reads were matched with novel miRNA in AE and HE sRNA libraries. The remaining sRNA reads were classified into other components, such as ribosomal RNA (rRNA), transfer RNA (tRNA), small nuclear RNA (snRNA), and small nucleolar RNA (snoRNA), as well as exon and intron regions of the gene (Figure 2B). Finally, the TPM density distribution of miRNA expression in AE and HE samples exhibited a similar distribution state (Figure 2C).

### 2.2. Correlation Analysis of AE and HE Samples and Differential Expression Analysis of miRNA

Principal component analysis (PCA) and correlation analysis were utilized to demonstrate the disparity between different biological replicates of the same state and various states’ differences. As illustrated in Figure 3A,B, samples belonging to the same group were clustered together, whereas Pearson correlation coefficients within the group were high. Different groups clustered in various areas, with significant differences.

To investigate spontaneous embryo loss-related miRNAs in the endometrium, 464 miRNAs were identified in AE and HE samples, including 357 known miRNAs and 107 novel miRNAs (Appendix A). Among 357 known miRNAs, 336 were co-expressed in AE and HE, while 7 and 14 miRNAs were specifically expressed in AE and HE, respectively (Figure 3C, Appendix A). For 107 novel miRNAs, 81 were co-expressed in the above two types of samples, while 16 and 10 miRNAs were specifically expressed in AE and HE, respectively (Figure 3D, Appendix A). The top 20 miRNAs with the highest expression levels in AE and HE are listed in Appendix A. We used *q*-value < 0.05 as the screening criterion for differentially expressed miRNAs (DEMs); 139 DEMs were found. Compared with HE samples, 66 miRNAs were highly expressed, and 73 miRNAs were low expressed in AE samples (Figure 4A). Hierarchical cluster analysis revealed the expression status of miRNA in AE and HE (Figure 4B). Finally, the top 20 DEMs of AE and HE samples were created (Appendix A).

### 2.3. Prediction of Target Genes of DEMs

The target genes were predicted by taking the intersection part of miRanda and RNAhybrid software prediction results, as shown in Appendix A. The mRNA sequencing data in the SRA database and miRNA sequencing data in this paper were jointly analyzed. The screening principles of miRNA–mRNA pairs were as follows: miRNA was significantly differentially expressed, mRNA was significantly differentially expressed, and miRNA and mRNA presented targeted interaction predicted by algorithms, with a Pearson correlation coefficient of <−0.8 between miRNA and mRNA. The interaction network of 139 miRNAs and their corresponding target genes was mapped. As illustrated in Figure 5, miR-671-5p, miR-885-3p, miR-365-5p, and miR-205 have many target genes and may regulate the expression of many genes, as shown in Appendix A, indicating that these miRNAs may be critical for post-implantation embryo development from the endometrial level.

### 2.4. Quantitative Real-Time PCR (qRT-PCR) Verified the Sequencing Results

To validate the accuracy of sequencing results, we selected eight DEMs and eight differentially expressed mRNAs to examine the sequencing results by qRT-PCR assay. Among the eight miRNAs validated above, miR-205, miR-365-5p, miR-671-5p, and miR-885-5p presented more target genes; miR-217, miR-503, miR-504, and miR-375 were obtained by random selection. Among the eight differentially expressed mRNAs, *TINAGL1* was the predicted target gene of miR-205, Hepatic and Glial Cell Adhesion Molecule (*HEPACAM)* was the predicted target gene of miR-365-5p and miR-671-5p, and T-Box Transcription Factor 6 (*TBX6)* was the predicted target gene of miR-671-5p and miR-885-3p. Their expression levels were determined using qRT-PCR (Figure 6). The final results showed that the qRT-PCR results were consistent with the sequencing results, which confirmed the sequencing results’ reliability.

### 2.5. Functional Enrichment of DEMs in Endometrial Tissue

To understand that miRNA regulates target genes and target genes regulate biological activities, possible differential expression target genes of miRNA were predicted to be enriched into Gene Ontology (GO) functional annotation and the Kyoto Encyclopedia of Genes and Genomes (KEGG) pathway enrichment based on hypergeometric tests. GO enrichment analysis showed that the target genes were mainly involved in tissue development, epithelium development, muscle structure development, morphogenesis of epithelium, epithelial tube morphogenesis, and regulation of nervous system development (Figure 7A; Appendix A). KEGG analysis showed that 262 pathways were enriched, the first 30 of which were listed (Figure 7B; Appendix A) and mainly involved in Antifolate resistance, Glycine, serine and threonine metabolism, folate biosynthesis, Pyrimidine metabolism, Serotonergic synapse, Arachidonic acid metabolism, and Circadian entrainment. At the same time, many reproductive-related pathways were enriched, such as steroid hormone biosynthesis, ovarian steroidogenesis, and the Estrogen signaling pathway.

### 2.6. miR-205 Directly Targets 3′ UTR of TINAGL1 to Reduce Its Expression

miR-205 is the top known high expression miRNA in AE compared to HE. The fluorescence in situ hybridization (FISH) assay was used to determine the localization of miR-205 in AE and HE samples, which was abundantly expressed elevated in AE luminal epithelium and glandular epithelium, and at the same time, slightly expressed in HE (Figure 8). It is consistent with our previous analysis of sequencing results and qRT-PCR results.

When mRNA sequencing data and target gene prediction software were combined, miR-205 was found to have many target genes, of which *TINAGL1* is linked to embryonic development and angiogenesis. Therefore, we demonstrated the combination of the seed region of miR-205 and 3′ UTR of *TINAGL1* using a dual-luciferase reporter gene assay system. The seed region of miRNA was entirely complementary with *TINAGL1* 3′ UTR but unmatched with a 3′ UTR mutation (Figure 9A). The plasmid construction results are depicted in Figure 9B.

To exclude the interaction between miR-205 and PGL3 plasmid, miR-205 mimic or miR-205 NC was co-transfected into 293T cells with the PGL3 NC. Co-transfection of miR-205 mimics into 293T cells with *TINAGL1* 3′ UTR WT plasmid resulted in a significant decrease in firefly/Renilla luciferase activity compared with co-transfection miR-205 NC into 293T cells with *TINAGL1* 3′ UTR WT plasmid group. Neither miR-205 mimic nor miR-205 NC was co-transfected into cells with *TINAGL1* 3′ UTR MT plasmid, and there were no significant differences in results between the two groups (Figure 9C). The above experiments revealed that miR-205 inhibited the *TINAGL1* gene expression by directly targeting 3′ UTR of the *TINAGL1* and the predicted targeted binding region was accurate.

## 3. Discussion

Since the discovery of the first miRNA lin-4, research on miRNA has recently grown in popularity. miRNA has been demonstrated to regulate the expression of about 20–30% of genes [22]. Research indicates that some miRNAs are correlated with germ cell development, embryonic development, mother–fetal exchange, and placentation [20,23,24]. This study used high-throughput sRNA sequencing technology to determine the RNA expression profile of AE and HE tissue on DG 28 to increase our understanding of the molecular mechanism in the endometrium involved in embryo development following implantation. Finally, 464 miRNAs were identified in AE and HE, including 357 known miRNAs and 107 novel miRNAs. These findings promote miR-205 to suppress endometrial angiogenesis by targeting *TINAGL1*, thereby inhibiting the post-implantation development of embryos.

Prior data reported that glucose transporter type 4 (*GLUT4)* in the endometrial epithelium affects embryo development [25]. When we conducted GO enrichment analysis on DEM’s differentially expressed target genes in AE compared with HE, we found that the biological process of significant enrichment is epithelial development. Previous studies showed that osteopontin expressed by uterine epithelium interacts with integrins on the placenta and affects embryo attachment and placentation in pigs [26].

KEGG pathway enrichment analysis can help us understand the molecular functions of DEMs by identifying the biological processes involved in the target genes corresponding to these DEMs. Some pathways enriched by these target genes are involved in reproductive processes, such as the NF-kappa B signaling pathway, Notch signaling pathway, Rap1 signaling pathway, and cell adhesion molecules. The following are some previous studies on these pathways. Han et al. demonstrated that after infecting porcine with circovirus type 2, the NF-kappa B signaling pathway could promote interleukin-1beta (IL-1β) and IL-10 anti-inflammatory cytokines in porcine alveolar macrophages, thereby killing germs [27]. In humans, the weakened Notch signal is associated with endometriosis. In addition, downregulating Forkhead box other 1 (*FOXO1*) expression eventually leads to decidua damage [28]. The Rap1 signaling pathway is a crucial immune-related pathway and cancer-related pathway [29,30]. The expression of cell adhesion molecules is critical for embryo implantation and pregnancy establishment [31]. Simultaneously, during the estrus cycle and DG 15 to 16, the most significantly enriched biological pathways of differentially expressed genes in the endometrium are cell adhesion molecules pathways [32]. These pathways enriched by target genes above indicate that miRNA regulates the expression of target genes and then influences corresponding molecular biological pathways.

The top three miRNAs highly expressed in AE and HE tissues are miR-21-5p, miR-148a-3p, and miR-143-3p. Studies demonstrated that miR-21-5p can promote Thp-1 cells and non-small cell lung cancer cell proliferation [33,34]. In another study, it could promote extracellular matrix degradation and angiogenesis [35,36]. miR-148a-3p can enhance the bactericidal and antibacterial ability of macrophages [37]. miR-143-3p has been demonstrated to inhibit the proliferation, migration, and invasion of ovarian cancer, hepatocellular carcinoma cells, endometriotic stromal cells, and osteosarcoma cells [38,39,40,41].

The top miRNAs with more targets are miR-365-5p, miR-671-5p, miR-885, and miR-205. Studies have shown that miR-365-5p can target RNA-binding protein with serine-rich domain 1 (*RNPS1)*, Keratin 14 (*KRT14)*, and DnaJ heat shock protein family (Hsp40) member B6 (*DNAJB6)* in porcine Alveolar Macrophages, thereby affecting interferon-mediated immune response [42]. Previous studies have shown that the differentially expressed miR-671-5p in the endometrium of Meishan and Duroc sows affects maternal placental development by targeting the Estrogen Receptor 1 (*ESR1)* gene [5,43]. Among the genes targeted by the above miRNAs. Deborah et al. showed that *TBX6* plays an important role in mesoderm specification in mouse embryos [44]. Anna-Katerina et al. showed that *TBX6* regulates left/right patterning in mouse embryos through effects on nodal cilia and perinodal signaling [45].

miR-205 regulates epithelial to mesenchymal transition by targeting Zinc-finger E-box Binding Homeobox 1 (*ZEB1*) and Smad Interacting protein 1 (*SIP1*), which helps the embryonic development process [46]. Previous studies have shown that the matricellular protein *TINAGL1* is a pro-angiogenic factor that plays a vital role in angiogenesis during pregnancy [47,48]. At the same time, studies have shown that *TINAGL1* supports the structure and function of blood vessels. *TINAGL1* is a crucial component of Reichert’s membrane, allowing gas and nutrient exchange between the maternal placenta and the embryo, supporting embryonic development after implantation [49,50].

Finally, *TINAGL1* plays an essential physical and biological role in mouse embryonic development [47,51]. Based on the two assays of the FISH and dual-luciferase report gene and the literature reports mentioned above, we speculated that in the AE, *TINAGL1* is targeted by its highly inversely related miR-205 to inhibit expression. *TINAGL1* is a promoter of angiogenesis, and finally, miR-205 inhibits maternal endometrium angiogenesis by inhibiting *TINAGL1*. However, more studies are required to confirm this conjecture.

## 4. Materials and Methods

### 4.1. Animal Sample Collection

All animals used in this study were approved by the Animal Care and Use Ethics Committee of South China Agricultural University (permit number: SYXK-2019-0136). The animal sample collection process was identical to our previously published paper [4]. Briefly, four Tibetan sows with similar weight and size (Parity 2) were selected. The first artificial insemination occurred immediately after the first estrous cycle and the second insemination occurred after 12 h. Sows were slaughtered at a local abattoir on DG 28 to obtain AE and HE samples in each sow’s uterus. Swine uterus samples were collected and shipped to the laboratory in an ice box, and then uterine samples were cut longitudinally from the anti-mesometrial side. According to generally accepted guidelines, we distinguish between healthy and arresting embryos based on embryo size, weight, and vascularity of the placental membranes. When the embryo is relatively large and heavy, the blood vessels of the placental membrane are abundant, it is considered to be a healthy embryo, and when the state of the embryo is reversed, we define it as arresting embryo [6,8,52]. After statistics, the ratio between healthy embryos and arresting embryos is about 3:1–4:1. After dissociating the embryos, we dissected the endometrium of the different embryo attachment sites as AE and HE, respectively [53]. On the one hand, uterine samples consisting of myometrium and endometrium were fixed in paraformaldehyde for the following paraffin section preparation and FISH histological observation. On the other hand, after collecting and labeling AE and HE samples and carefully transferring them, the samples were first frozen in liquid nitrogen and subsequently transferred to a freezer at −80 °C for long-term preservation for RNA extraction.

### 4.2. Library Preparation for sRNA Sequencing

All RNAs were extracted from endometrial tissue samples using TRIzol reagent (Invitrogen, Carlsbad, CA, USA) following the manufacturer’s instructions. A 1% agarose gel assay confirmed the lack of decomposition and contamination of the extracted RNA. A NanoPhotometer^®^ spectrophotometer measured RNA purity at 260 and 280 nm (IMPLEN, Westlake Village, CA, USA). Accurate RNA concentrations were measured using the Qubit^®^ RNA Assay Kit in Qubit^®^ 2.0 Fluorometer (Life Technologies, Carlsbad, CA, USA). RNA integrity number (RIN) was measured using the RNA Nano 6000 Assay Kit of the Agilent Bioanalyzer 2100 system (Agilent Technologies, Santa Clara, CA, USA). RIN values exceeded 7.60.

A total of 3 μg RNA per sample were used as the basis for small RNA library construction. The sequencing library was performed using NEBNext^®^ Multiplex Small RNA Library Prep Set for Illumina^®^ (NEB, Ipswich, MA, USA). First, index codes were added to each sample as a sample property. The NEB 3′ SR Adaptor was ligated to the 3′ end of the miRNA, and the SR RT Primer hybridized with the excess of 3′ SR adaptor, converting the single-stranded DNA Adaptor into a double-stranded DNA molecule, thereby inhibiting adaptor dimer formation. This double-stranded DNA was not a substrate mediated by T4 RNA Ligase 1 and, therefore, could not be attached to the 5′ SR adaptor in the subsequent ligation reaction. At the same time, the 5′ end adaptor was ligated to the 5′ end of the miRNA, the first-strand cDNA was synthesized using M-MuLV reverse transcriptase (NEB, Ipswich, MA, USA), and then the PCR process was conducted by use of LongAmp Taq 2X Master Mix (NEB, Ipswitch, MA, USA), SR primer, and index primer. The PCR product was purified by 8% agarose gel, and the DNA fragment of 140–160 bp (miRNA length and the length of 3′ adapter and 5′ adapter) was recovered by cutting the gel and placed in 8 μL of elution buffer.

After library creation, the sRNA library was diluted to 1 ng/µL. DNA High Sensitivity Chips determined library quality on the Agilent Bioanalyzer 2100 system (Agilent Technologies, Santa Clara, CA, USA). Q-PCR was performed to detect the exact concentration of constructed library. Finally, the sRNA library was sequenced at Illumina HiSeq 2500 platform (Illumina, San Diego, CA, USA).

### 4.3. Sequencing Data Analysis

The raw reads obtained by sequencing are subjected to the quality control described below. Error rate, Q20, Q30, and GC-content of each sample’s raw reads sequencing data were calculated.

By removing reads with an N proportion more significant than 10%, with 5′ adapter contaminants, without 3′ adapter or the insert tag, having to ploy A/T/G/C, and low-quality reads, raw reads can be converted into clean reads. Using length screening treatment, clean reads can be further screened. Generally, animal sRNA length ranges from 18 to 35 nt, and sRNA length distribution can help identify the sRNA category.

After length screening, the clean reads were compared to the reference sequence using Bowtie (v.0.12.9), and their expression level was analyzed [54]. miRBase20.0 was employed as a miRNA reference database [55]. mirDeep2 (v.2.0.0.5) was utilized to match known miRNA. The available software miREvo (v.1.1) and mirDeep2 (v.2.0.0.5) were comprehensively used to predict novel miRNA [55,56]. Reads were classified as miRNA, rRNA, tRNA, snRNA, snoRNA, repeat, novel miRNA, exon, intron, and others (Appendix A).

### 4.4. Identification of DEMs

To eliminate the effect of sequencing depth and RNA length on read counts, the read count of miRNA was normalized into transcripts per million (TPM) for the next screening of DEMs [57]. Differential expression analysis was performed using the DESeq R package (v.1.8.3) on AE and HE libraries to obtain each miRNA’s fold change and *p*-value [58]. The Benjamini–Hochberg method was used to convert the original *p*-value to a *q*-value to improve the analysis’s reliability [59,60]. A *q*-value < 0.05 was defined as the threshold of significant differential expression. This experiment included four biological replicates in AE and HE, respectively. The results of differential expression analysis and cluster analysis were represented by a volcano diagram and hierarchical clustering heatmap.

### 4.5. Target Gene Prediction

The target genes of DEMs were predicted using miRanda (v.3.3) and RNAhybrid (v.2.0) software [61,62]. Only once the target gene was identified in both software was it is considered the final target gene of miRNA. DEMs in this experiment were combined with mRNA sequencing data uploaded to the SRA database by Zang et al. [53]. We obtained the intersection of the following four data sets: Pearson correlation coefficients between miRNA and mRNA < −0.8, DEMs, differentially expressed mRNAs, miRNAs and mRNAs were targeted interactions (Appendix A), and finally, network interaction diagrams were constructed using Cytoscape (v.3.9.1) software to illustrate the interrelationships between miRNAs and mRNAs [63].

### 4.6. Functional Analysis of DEMs

To elucidate the biological functions and pathways underlying DEMs, we conducted GO and KEGG pathway analyses on target genes predicted by DEMs. GO is the standard of gene functional classification, and KEGG is a database integrating genomic, chemical, and systemic functions [64]. GO analysis mainly included molecular function, biological process, and cellular component. Rich factor and *q*-value were utilized as parameters to select the top 30 enriched pathway terms to display in KEGG pathway analyses.

### 4.7. Validation of miRNA and mRNA Expression via qRT-PCR

Using qRT-PCR methods, eight samples from AE and HE groups were used to verify the accuracy of sRNA and mRNA sequencing data [65]. Three biological replicates were performed in two technical replicates [66]. Eight DEMs and eight mRNA were selected for qRT-PCR. The miRNA and mRNA primer sequences are listed in Appendix A. All miRNA in the sample was reverse transcribed into cDNA using the PolyA RT-PCR method, miRNA reverse transcription reagents were provided by the Mir-X miRNA First-Strand Synthesis Kit (Takara, Dalian, China), and mRNA reverse transcription reagents were supplied by the PrimeScriptTM RT reagent Kit with gDNA Eraser (Takara, Dalian, China); all reagents were used following the manufacturer’s instructions. We performed qRT-PCR in the Applied Biosystems^®^ QuantStudio™ 7 Flex Real-Time PCR System (Thermo Fisher, Singapore) using PowerUp™ SYBR™ Green Master Mix (Thermo Fisher, Vilnius, Lithuania). As previously studied, *U6* and Hypoxanthine Phosphoribosyltransferase 1 (*HPRT1*) were used as internal reference genes for miRNA and mRNA, respectively [67,68]; meanwhile, according to the previous miRNA and mRNA sequencing data, *U6* and *HPRT1* were stably expressed in AE and HE with similar expression levels. In summary, we selected the above two genes as internal reference genes. The relative expression level of miRNA and mRNA was calculated using a comparative cycle threshold (2^−^^∆∆^^Ct^).

### 4.8. FISH Assay

The FISH assay was used to detect the localization and relative quantification of miR-205 in endometrial tissue; the probe sequence of miR-205 is 5′-AGGAAGTAAGGTGGCCTCAGAC-3′. The basic process was as follows. The clean tissue was placed in 4% paraformaldehyde (Servicebio, Wuhan, China) fixed fluid for more than 12 h, then dehydrated with graded ethanol (SCRC, Shanghai, China), and then embedded in paraffin. After the paraffin was sliced with a microtome (Leica, Shanghai, China), the sections were extracted using a water bath slide (Kedee, Jinhua, China) and incubated for 2 h at 62 °C. The paraffin sections were placed in BioDewax and Clear solution (Servicebio, Wuhan, China) for 15 min and were dehydrated twice in pure ethanol for 5 min. Then, they were dehydrated in 85% and 75% graded ethanol for 5 min each and washed in DEPC (Amresco, Solon, Ohio, USA) dilution. The sections were boiled in the retrieval solution for 15 min and allowed to cool naturally. Liquid Blocker PAP Pen (Servicebio, Wuhan, China) was used for tissue labeling, and Proteinase K (20 μg/mL) was used for digestion. After rinsing with pure water, they were rinsed three times with phosphate-buffered saline (PBS). Then, pre-hybridization solution was added dropwise to the sections and incubated for 1 h at 37 °C. The pre-hybridization solution was removed, the probe-containing hybridization buffer (Servicebio, Wuhan, China) was added dropwise, and the hybridization was performed overnight in a 37 °C incubator (labotery, Tianjin, China). Then, the hybridization solution was removed, washed with 2 × SSC (Servicebio, Wuhan, China) solution for 10 min, washed twice with 1 × SSC solution for 5 min each, and washed with 0.5 × SSC solution for 10 min. Next, DAPI (Servicebio, Wuhan, China) staining solution was added dropwise to the sections, incubated in the dark for 8 min, and anti-fluorescence quenching sealing tablets (Servicebio, Wuhan, China) were added dropwise after rinsing. Finally, the sections were observed under a fluorescence microscope (Nikon, Tokyo, Japan), and images were collected.

### 4.9. Dual-Luciferase Report Gene Assay

*TINAGL1* 3′ UTR wide-type (WT) and *TINAGL1* 3′ UTR mutant (MT) were constructed downstream of the firefly luciferase reporter gene in the pGL3 promoter vector, resulting in *TINAGL1* 3′ UTR WT plasmid and *TINAGL1* 3′ UTR MT plasmid, with the original PGL3 promoter vector as a PGL3 negative control (NC). Considering that 293T cells have a higher transfection efficiency, they were used as target cells for transfection. After cell resuscitation, 293T cells were subcultured and inoculated into 96-well plates with 10,000 cells per well. Finally, we co-transfected the three types of plasmids into 293T cells with miR-205 mimic or miR-205 negative control (NC) according to the manufacturer’s instructions. At the same time, we did not transfect the plasmid and miRNA, but only cells and transfection reagent were kept as a negative control group. Three technical repetitions were conducted per group.

After 48 h of co-culture, luciferin and coelenterazine were added to the cells using a dual-luciferase reporter gene assay kit (Beyotime Biotechnology, Shanghai, China) and following the manufacturer’s instructions. Finally, we used an Infinite M100 Pro (TECAN, Grödig, Austria) microplate reader to detect the numerical size of luciferase and perform further statistical analysis to calculate the ratio of firefly luciferase values to Renilla luciferase values.

### 4.10. Statistical Analysis

The *t*-test was employed to determine the significance of qRT-PCR and dual-luciferase reporter gene assay using GraphPad Prism software (v.9.3.1). A *p*-value < 0.05 was considered statistically significant.

## 5. Conclusions

sRNA sequencing technology determined miRNAs of AE and HE on DG 28 in pigs. A total of 357 known miRNAs and 107 novel miRNAs were identified. After comparing AE and HE tissues, 139 DEMs were identified (66 high expression and 73 low expression). The GO analysis indicates that these DEMs may affect embryonic development by regulating biological processes related to epithelial development and amino acid metabolism. The results of differential expression analysis of miRNA and published differentially expressed mRNA expression profiles were analyzed together. The interaction network between miRNA and target genes, GO function, and KEGG pathway analysis was performed to determine the molecular association between miRNA, mRNA, and post-implantation embryo development from the endometrial level. Finally, our findings help us to gain a better understanding of the role of miRNAs in the regulation of embryonic development following implantation and embryonic survival in pigs.

## Figures and Tables

**Figure 1 ijms-23-08157-f001:**
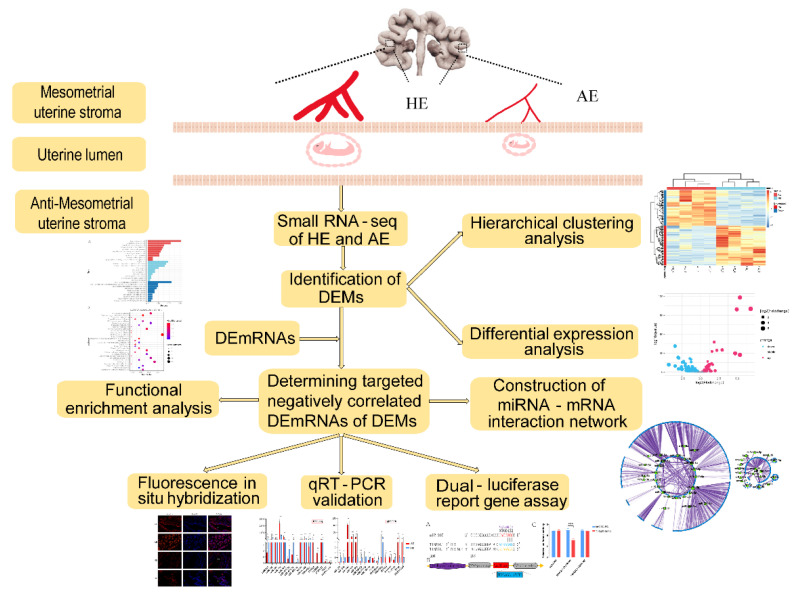
This flow chart displays the design ideas and experimental and analytical procedures of this study. The specific details are shown in the text.

**Figure 2 ijms-23-08157-f002:**
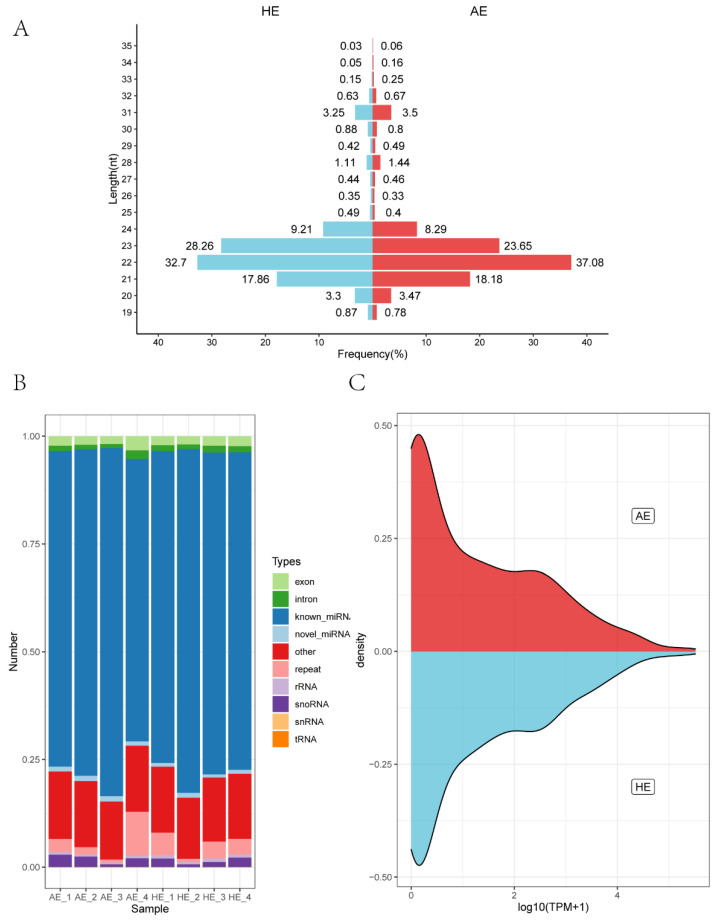
Summary of small RNA (sRNA) sequencing results of 8 samples from two states. (**A**) sRNA length distribution, miRNA mainly distributed in 21–23 nt, in which 22 nt accounted for the highest proportion. (**B**) sRNA classifications in 8 samples from arresting endometrium (AE) and healthy endometrium (HE) were annotated into the reference sequence. (**C**) The density distribution of miRNA expression.

**Figure 3 ijms-23-08157-f003:**
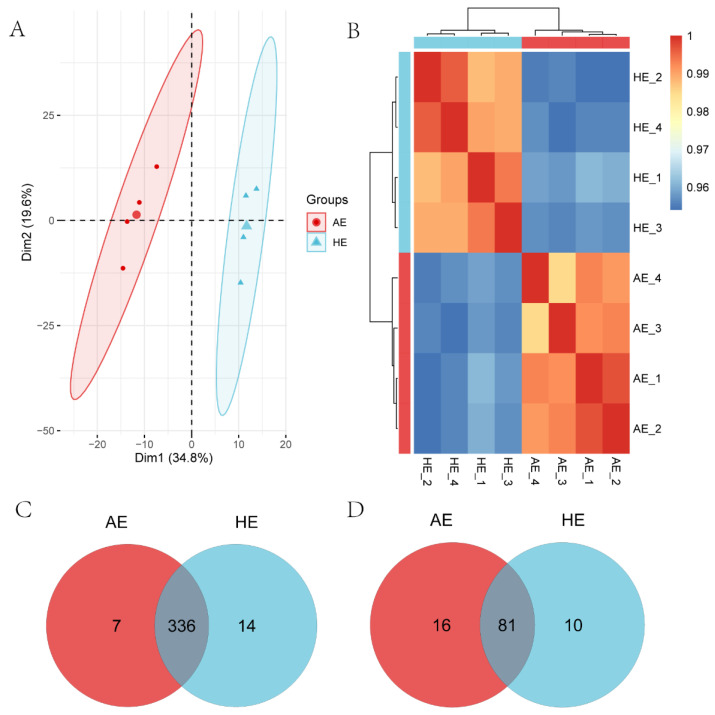
Correlation and miRNA expression levels in porcine endometrial samples. (**A**) Principal component analysis (PCA) between arresting endometrium (AE) and healthy endometrium (HE) samples. (**B**) Pearson correlation coefficients between samples show the correlation. The color scale is ramped up from 0.95 (blue, low correlation) to 1.00 (red, high correlation). (**C**) Venn diagrams of 357 known miRNAs in AE and HE samples. (**D**) Venn diagrams of 107 novel miRNAs in AE and HE samples.

**Figure 4 ijms-23-08157-f004:**
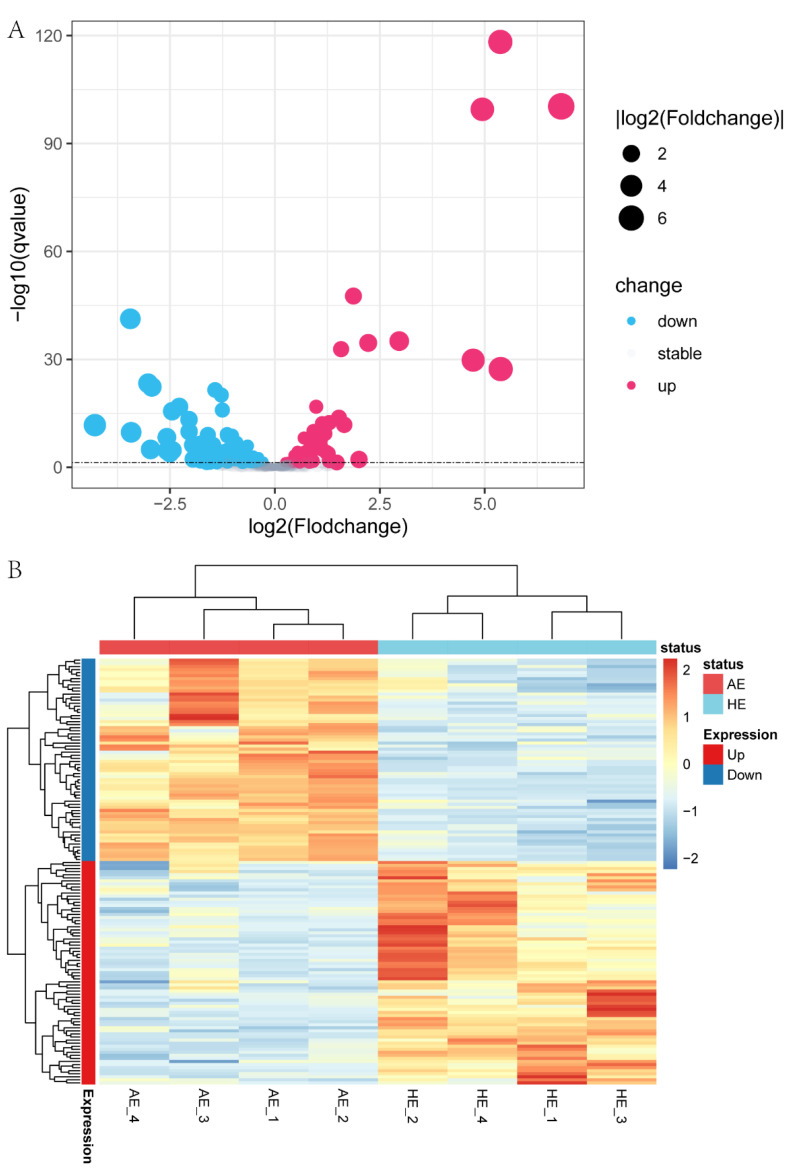
Differential expression of miRNA in arresting endometrium (AE) and healthy endometrium (HE) samples. (**A**) Each dot represents one miRNA for the differentially expressed miRNAs (DEMs) in the porcine endometrium. The drop size represents the fold change gradient of DEMs expression level. The three different colors represent upregulation, downregulation, and stable expression. (**B**) The hierarchical clustering heatmap showed DEMs expression status in AE and HE. The color scale gradually increases from −2 (blue, low miRNA expression level) to 2 (red, high miRNA expression level).

**Figure 5 ijms-23-08157-f005:**
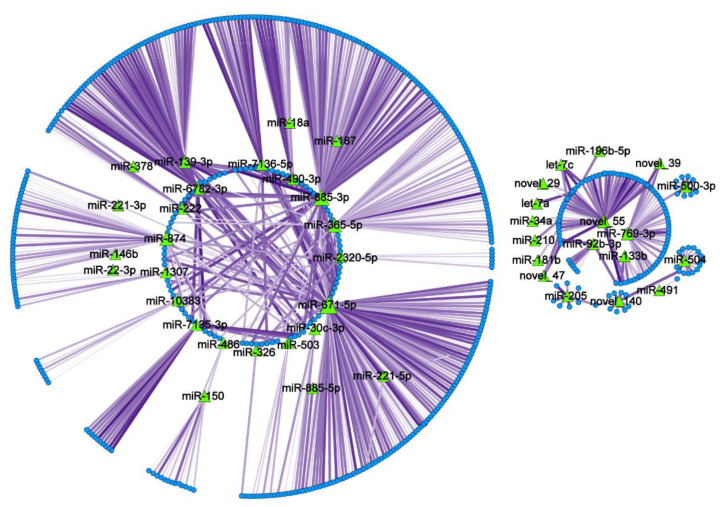
miRNA-mRNA interaction network diagram. Green triangles represent miRNAs, blue dots represent mRNAs, lines between miRNA–mRNA indicate interaction and the purple gradient of the line represents the absolute value of the correlation coefficient (dark color—large correlation coefficient, strong correlation. light color—small correlation coefficient, weak correlation), and the thickness of the line represents the *p*-value for predicting the significance of the miRNA–mRNA negative correlation (thick, the difference is extremely significant, thin, significant difference).

**Figure 6 ijms-23-08157-f006:**
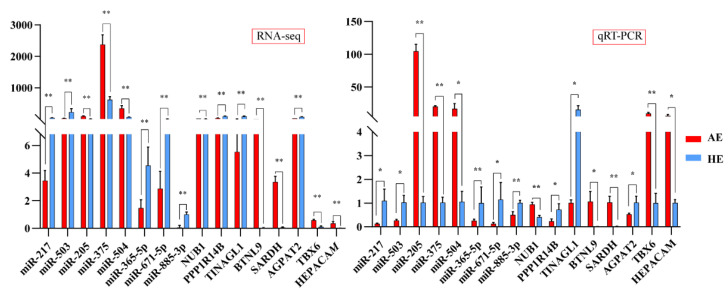
Validation of the expression of miRNAs and mRNAs using quantitative real-time PCR (qRT-PCR). The left and right panels represent the data of RNA-seq and qRT-PCR, respectively. The red bar represents AE, the blue bar represents HE, mean ± the standard deviation (SD) denotes the results, * represents *p*-value < 0.05, ** represents *p*-value < 0.01.

**Figure 7 ijms-23-08157-f007:**
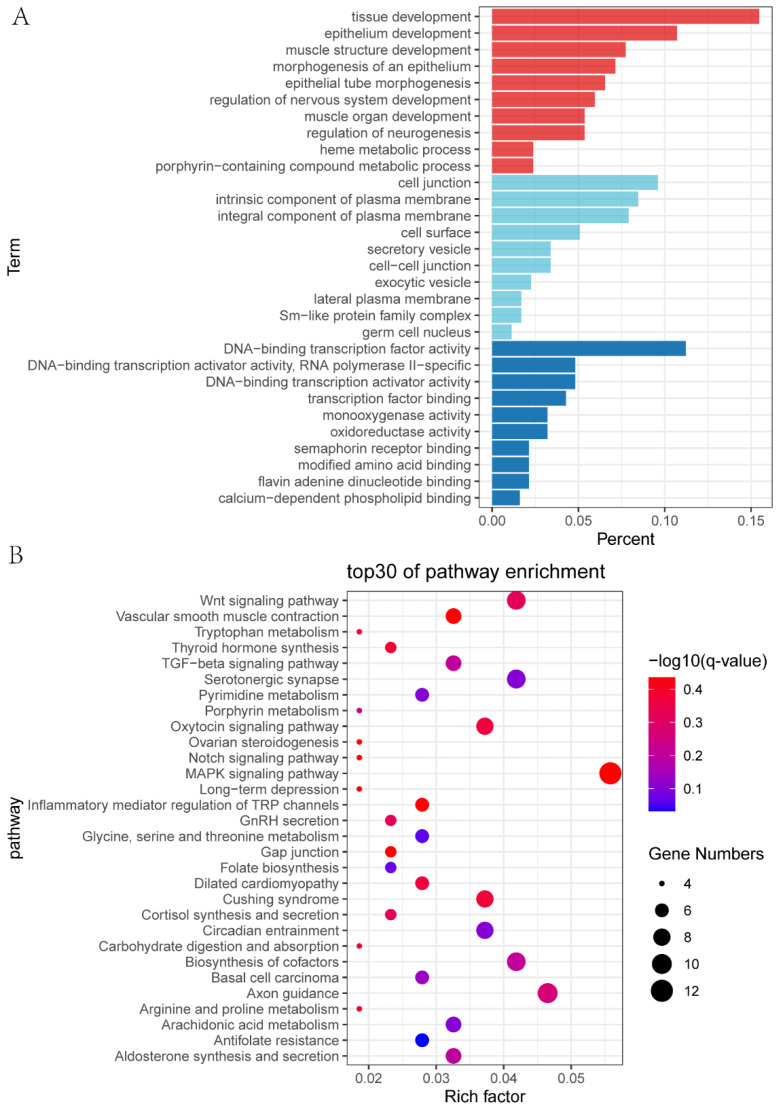
Functional enrichment of differentially expressed target genes of differentially expressed miRNAs (DEMs). (**A**) Gene Ontology (GO) enrichment results were divided into three categories: red bars represent biological processes, light blue bars represent cellular components, and dark blue bars represent molecular functions. There are 10 GO terms for each type. (**B**) The top 30 terms in the Kyoto Encyclopedia of Genes and Genomes (KEGG) pathway analysis.

**Figure 8 ijms-23-08157-f008:**
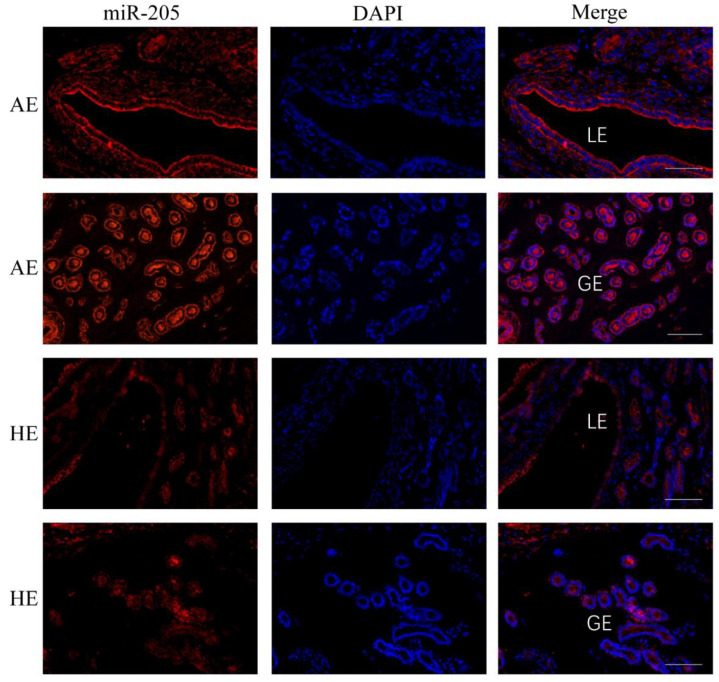
Fluorescence in situ hybridization (FISH) localized expression of miR-205 to the uterine section samples in the two statuses of arresting endometrium (AE) and healthy endometrium (HE). At 28 days of pregnancy, the expression level of miR-205 was abundantly expressed in AE luminal epithelium and glandular epithelium, and at the same time, it was slightly expressed in HE tissues. Hybridization buffer without probes was used as a negative control group (NC; (Appendix A)). Legend: LE, luminal epithelium; GE, glandular epithelium. Scale bar 100 um.

**Figure 9 ijms-23-08157-f009:**
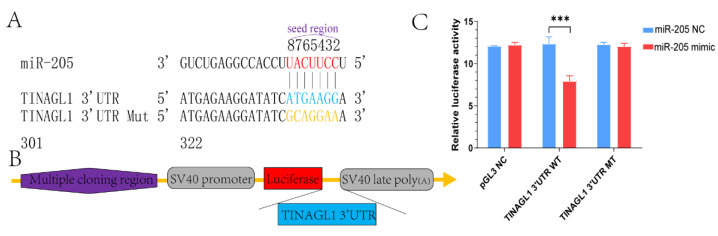
miR-205 targets the 3′-untranslated regions (3′ UTR) of tubulointerstitial nephritis antigen-like 1 (*TINAGL1*). (**A**) miR-205 seed sequence and *TINAGL1*′s 3′ UTR predicted complementary pairing sites. (**B**) *TINAGL1* 3′ UTR wide type (WT) was constructed downstream of the firefly luciferase reporter gene in the pGL3 promoter vector. (**C**) miR-205 mimic or miR-205 negative control (NC) was co-transfected into 293T cells with the PGL3 negative control (NC), *TINAGL1* 3′ UTR WT plasmid, and *TINAGL1* 3′ UTR mutant (MT) plasmid. In the groups that co-transfected miR-205 mimic and *TINAGL1* 3′ UTR wide type, the firefly/Renilla luciferase activity was significantly decreased compared with the control group. Demonstrating that miR-205 may target *TINAGL1* to affect luciferase activity. Data are presented as mean ± the standard error of the mean (SEM), *** represents *p*-value < 0.001.

## Data Availability

The relevant datasets for this study are stored in an online database. The raw data used in this study are stored in the Sequence Read Archive (SRA) of NCBI, accession number PRJNA810419.

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
