# Peer review of "Differential MicroRNA Expression in Porcine Endometrium Related to Spontaneous Embryo Loss during Early Pregnancy"

_ijms, 2022, doi:10.3390/ijms23158157_

Round 1

Reviewer 1 Report

The reviewed paper presents transcriptomic data on miRNA expression in the arresting and healthy porcine endometrium on day 28 of pregnancy and have been analyzed with the use of bioinformatics tools. The obtained data was validated by quantitative real-time PCR. The authors chose miR-205 for detailed studies. They confirmed its expression in the porcine endometrium (FISH) and its participation in the inhibition TINAGL1 gene expression (Dual-Luciferase Report Gene Assay), which is related to angiogenesis.

In general, the presented data is interesting, however, the method of its presentation is not well-developed and the manuscript needs linguistic correction.

Detailed remarks:

ABSTRACT AND INTRODUCTION

Lines 25-26: it should be “… of porcine endometrium on day (DG) 28 of gestation.”

Line 35: The name of the gene should be in italics.

Lines 38-37: The summary of the work should be more precise. The general statement that the publication “provides a scientific reference for promoting successful implantation of embryos in early pregnancy” is in my opinion insufficient.

Lines 48-49: I propose to add: “… is abnormalities in mother-fetal communication”

Lines 49-51: This sentence requires correction, because the term “to fully coordinate mutual” is not appropriate in my opinion. For example: “These interactions are controlled by exchange of signals between the embryo and endometrium, such as: estrogen, miRNAs, extracellular vesicles, cytokines, chemokines, growth factors, mRNA destabilizing factors, and other substances.”

Lines 57-58: It should be clearly indicated the difference between sRNA and miRNA. Moreover it should be noted that miRNA is a subgroup of sRNA. This will help in understand the rest of the manuscript.

Line 65: the term “For instance, …”, does not fit here, it should be for example: “It has been shown that …”.

Lines 74-76: This sentence is too long, it should be split. In addition, more specific information on the TINAGL1 gene should be added. The name of the gene should be in italics.

Lines 83-84: remove “and” before “specific details are shown in the text.”.

RESULTS

Line 87: It should be clarified whether 8 samples were used for the entire experiment or 8 samples per group.

Lines 91-92: Table numbering should be kept in the order. The referenced table should be marked as: Table S1. This comment applies to all tables in the text.

Line 96: “As a result, miRNA length screening was conducted (Figure 2A).” - this sentence is incorrectly worded. I propose to correct them, for example: “Therefore a miRNA length screening was performed (Figure 2A).”

Lines 109-110: I propose to change this sentence: “The remaining sRNA reads were classified into other components, such as rRNA, tRNA, snRNA, and snoRNA, as well as exon and intron regions of the gene (Figure 2B).”.

Lines 145-147: The description of the figure should include information that the presented data were obtained by comparing AE and HE samples. In addition, three states of expression are mentioned in the legend: up-regulation, down-regulation, and stable expression; however, there are only two states (up-regulation and down-regulation) in the chart. This should be clarified or the graph should be modified.

Lines 163-166: I propose to attach to the Supplementary Materials a table with genes that are regulated by the listed miRNA.

Figure 5 – In my opinion bolding miRNA names will make the chart easier to read.

Lines 168-173: I suggest changing the word “deep” to “dark color” and “shallow” to “light color”. In addition, he suggests using a dash in a sentence after: “dark color”, “light color”, “thick” and “thin”.

Lines 174-185: The qRT-PCR validation results are not clearly described. At the beginning, it is indicated that 8 DEMs and 8 mRNAs have been validated, then 4 miRNAs are listed for which 3 target genes were selected, and then the information appears that: “The remaining 4 miRNAs and 5 mRNAs were obtained by randomization.”. Please explain this inaccuracy. In addition, gene names should be written in italics - this comment applies to the entire manuscript.

Figure 6 - One more division range should be added to the graph - not all graphs show the standard deviation.

Lines 187-190: It should be indicated whether the results are presented as mean with SEM or as median with SD.

Lines 194-195: “enrichment through an algorithm ” - what algorithm?

Lines 196-205: Process names should not be listed in quotation marks.

Lines 213-217 and 236-241: By using only FISH assay without any statistical analysis you cannot write that miR-205 expression was significantly elevated in AE luminal epithelium and glandular epithelium. It can only be seen from the staining that the signal is intense or weak.

Lines 231-234: “inhibited the translation expression” - what does it mean? I think it should be: “inhibited the TINAGL1 gene expression”.

Lines 231-234: It should be: “Data are presented as means ± the standard error of the mean (SEM), * * * represents p-value < 0.001.”.

DISCUSION

Lines 258-260: This sentence is not fully understood. Maybe it would be better to write: “This study used high-throughput sRNA sequencing technology to determine the RNA expression profile of AE and HE tissue on GD28 to increase our understanding of the molecular mechanism in the endometrium involved in embryo development following implantation.”

Lines 261-262: Two sentences start with “finally”. It should be corrected.

Lines 262-265: This sentence should be split into 2 or 3 separate sentences. Moreover, the use of the expression "hoping to provide a scientific reference and theoretical basis" is not accurate, it should be corrected.

Lines 266-267: Why do you only in this sentence use “ssc” before the miRNA name. In the rest of the manuscript only miRNA names alone are given.

Line 274: “Epithelial development plays a positive role in embryonic development” - this information must be extended and the source must be added.

Line 282: “Previous studies have shown that TINAGL1 is a pro-angiogenic factor that plays a vital role in angiogenesis during pregnancy” - it should be specified that this concerns the TINAGL1 protein.

Lines 292-294: this sentence should not begin with “In contrast”.

Comment to the discussion: The discussion is too short and does not provide information on the genes and miRNAs selected for validation (are they important during early pregnancy and implantation?). Only the information for the TINAGL1 gene is given.

MATERIAL AND METHODS

Lines 299-300: “(Parity 2)” - What does this mean? Are you sure this term “were screened”? I think it should be "selected".

Line 302: It should be: “Sows were slaughtered at a local abattoir.”. Moreover, the number of animals from which the samples was obtained should be given.

Lines 306-308: “When the embryo is relatively large and heavy, the blood vessels of the placental membrane are abundant, it is considered to be a healthy embryo, and when the state of the embryo is reversed, we define it as arresting embryo” - the weight range for healthy and arresting embryos should be given. Moreover, it must be determined whether each pig used in the experiment had any arresting embryos.

Lines 312-317: Two sentences start with “On the one hand”. It should be corrected.

Lines 342-344: “DNA High Sensitivity Chips determined library quality in the RNA Nano 6000 Assay Kit of the Agilent Bioanalyzer 2100 system” - is this information correct? First it talks about the DNA kit, then the RNA kit. Is this not a mistake?

Line 389: “Rich factor” - is this term correct?

Line 416: instead of "slicer" there should be "microtome”.

Lines 416-418: “oven roast” - it is not a scientific term. Just write that: “incubated for 2h at 62 °C.”.

Lines 425-426: Remove the "and" at the beginning of the sentence.

Lines 426-429: At what temperature was hybridization performed?

CONCLUSIONS

Lines 460-461: “(66 high and 73 low)” - it should be added that it is about expression.

Lines 463-464: instead “profiles were jointly analyzed” it should be “profiles have been analyzed together”.

In my opinion, this paragraph is not a conclusion from the work, but a summary of it. The conclusions of the work should be more specific. Sentence “DEMs and mRNA identified in this study establish a vital link for further investigation of the molecular mechanism of embryonic development following embryo implantation.” it is too general.

Reviewer 2 Report

The manuscript presents sRNA-seq to determine miRNAs of arresting endometrium (AE) and healthy endometrium (HE) on the 28th gestation day (GD 28) in pigs. A total of 357 known miRNAs and 107 novel miRNAs were detected in both the AE and HE endometrium tissues. In this study the Authors have focused on the role of miR-205, which is implicated in the physiology of the epithelia through the regulation of a variety of pathways that are related to differentiation and morphogenesis. It appeared that for AE miR-205 could inhibit its expression by combining 3’-untranslated 34 regions (3’ UTR) of tubulointerstitial nephritis antigen-like 1 (TINAGL1). In my opinion, the study presented by the authors is well designed, well written and is interesting since it contributes additional knowledge about the molecular mechanism that could be responsible for early embryonic death during the first month of pregnancy in the pigs. I suggest minor revision to improve the quality of the manuscript.
1. Improve the objective (L77-81). Provide the full names for abbrev/acronyms used for the first time in the text.

2. Re-check the presentation of Figures 3A-C. Figure 3C is presented after Figures 4A-B, making it a bit too confusing.

3. Add a concluding statement about the potential role of miR-205 in AE.

4. Give the period when the sows were slaughtered (L302).
